

# 1 Global long-term trends in the total electron content

Jaroslav Urbář, and Jan Laštovička
Institute of Atmospheric Physics, Czech Acad. Sci., Bocni II 1401, 14100 Prague, Czech Republic
*Correspondence:* Jan Laštovička (jla@ufa.cas.cz)
**Abstract**. The total electron content (TEC) is an important parameter for the ionospheric dynamics, GNSS/GPS
signal propagation and related applications of GNSS/GPS signals. Despite this fact the long-term trends in TEC
have been studied a little only. Here we analyze the homogeneous series JPL-35 of global TEC data for 1994-
2014 for selection of the optimum solar activity proxy for TEC analyses, and the UPC TEC data over 2003-2023
for estimating long-term trends in TEC. TEC trends are very predominantly negative. TEC trends reveal a clear
wavenumber 2 longitudinal structure in low/equatorial latitudes with strong negative trends in belts 0-60$^{\circ}$E and
180-240$^{\circ}$E and weak trends in 90-150$^{\circ}$E and 270-330$^{\circ}$E. For more detailed information on TEC trends a longer
series of reliable TEC data is required.

## 1 Introduction

The increasing atmospheric concentration of greenhouse gases, particularly of carbon dioxide, and long-term
changes of other trend drivers, mainly the secular change of the Earth magnetic field and of the stratospheric
ozone concentration, result in long-term trends in the thermosphere and ionosphere (e.g., Lastovicka et al.,
2012). Since the pioneering work by Rishbeth and Roble (1992) the investigations of long-term trends in the
ionosphere have been developing for more than 30 years. The state of investigations of long-term trends in the
mesosphere-thermosphere-ionosphere system has recently been reviewed by Lastovicka (2023).
One of the most important ionospheric parameters is the total vertical columnar electron content  (TEC),
particularly due to its impact on propagation of signals of the Global Navigation Satellite Systems (GNSS) such
as the Global Positioning System (GPS) and their applications, e.g. precise positioning, causing serious issues for
the single-frequency receiver-based positioning and for precise positioning using differential GNSS techniques,
like (Network) Real Time Kinematic (RTK/NRTK) (Hernández-Pajares et al., 2017). Global TEC data are
available only since 1994; therefore trends in TEC have been studied less than trends in other main ionospheric
parameters observed by the global ionosonde network since the International Geophysical Year in 1957/58. The
first paper on trends in TEC was published by Lean et al. (2011) for the period 1995-2010. They found the
average trend to be positive, which is not consistent with trends in foF2. Lastovicka (2013) used historical (1976-
1996) Faraday rotation-based TEC data from Florence, Italy, the region where Lean et al. (2011) trends were
positive and much stronger than average trends. He found no long-term trend but with relatively large
uncertainty, which however questioned results of Lean et al. (2011). Lean et al. (2016) analyzed TEC data over
the period 1999-2015 and obtained a very weak, statistically insignificant global TEC trend (negative but close





to zero). Emmert et al. (2017) constructed homogeneous TEC data series JPL-35 based on 35 globally
distributed stations re-evaluated consistently by the same method. They compared the evolution of JPL-35 data
with other data series for 1994-2014. Emmert et al. (2017, their Fig. 7) found non-stable level of TEC in early
years, particularly jump up of CODE data series by 3 TECU in autumn 2001. Lastovicka et al. (2017) used
Emmert's JPL-35 global TEC data series and found slight negative trend in global TEC and provided evidence
that the Lean (2011) positive trend was a consequence of data problem in early years (before autumn 2001) of
TEC data series, and "better" result of Lean et al. (2016) is due to the fact that they included less "wrong" years
into analysis.

Before studying TEC trends we have to solve the problem of optimum solar activity proxy for removal the solar
cycle effect, because for foF2 it was found that trends are critically dependent on selection of the optimum solar
activity proxy (Lastovicka, 2024). This is the first task of this paper. F30 was found to be the optimum solar
activity proxy for foF2 (Lastovicka and Buresova, 2023; Danilov and Berbeneva, 2023, 2024; Zossi et al., 2023).
The main task of this paper is to establish the regional TEC long-term trends globally.

In this work we shall examine the regional TEC trends globally in dependence on latitude and longitude over the
globe. Section 2 describes data and methods used. Section 3 deals with the selection of optimum solar activity
proxy for TEC investigations. Section 4 treats long-term trends in TEC. Section 5 contains conclusions.


**2 Data and methods**

To reach the first goal, to select the optimum solar activity proxy, the Emmert's et al. (2017) homogeneous
global average TEC data JPL-35 will be used (Emmert et al., 2017, Supplement, Data Set S1). We shall analyze
yearly average values based on monthly medians over the period 1994-2014. Criteria used for selection of the
optimum solar activity proxy are described in section 3.

To study the regional long-term trends in TEC, the UPC TEC global map data are used (Hernandez-Pajares et
al., 1998). We analyze yearly averages based on monthly medians around noon (10-14 local time) for 2003-
2023. The time interval has been selected to avoid data problems. Before 2002 the TEC data from all
international resources (IGS, CODE, JPL, UPC, ESA) are more or less unstable according to Emmert et al.
(2017), whereas since 2002 they are stable with respect to JPL-35. Moreover, UPC data were issued with epoch
having time step of two hours, before 2003 in odd hours and since 2003 in even hours. Data are separated by two
hours in local time (LT). Therefore to have all data in the same LT, we are performing analysis for averages in
meridional belts thick $30^o$ of longitude (equal to 2 hours of LT) with latitudinal step/resolution $2.5^o$. The first belt
is centered at $0^o$E, next at $30^o$E etc.

The long-term trends are calculated in the traditional way. First the effect of solar activity is removed from TEC
data in order to remove the much stronger solar cycle effect. Then the trends are calculated from TEC residuals
in the following way:



First, the dependence of TEC on solar proxies (i.e. parameters A and B) is calculated by linear regression, Eq.

81  (1):

$\quad\quad$ TEC = A + B * solar proxy $\quad\quad\quad\quad\quad\quad\quad\quad\quad\quad\quad\quad$ (1)
Second, using Eq. (1) with parameters A and B calculated in the first step, model values of $TEC_{mod}$ are
calculated for all individual years and all solar proxies. Third, using linear regression for TEC residuals $TEC_{obs}$ –
$TEC_{mod}$, Eq. (2):
$\quad\quad$ $TEC_{obs}$ – $TEC_{mod}$ = C + D * time $\quad\quad\quad\quad\quad\quad\quad\quad$ (2)
where $TEC_{obs}$ is the observed value of TEC, the long-term trend represented by the trend coefficient D is
calculated.


**3  Selection of the optimum solar activity proxy for TEC**

For the selection of the optimum solar activity proxy we use Emmert's (2017) homogenized TEC data JPL-35,
1994-2014, and six solar activity indices/proxies, F10.7, F30, Mg II, He II, sunspot number and the solar Lyman-
α flux. The optimum solar activity proxy selection requires criteria according to which the selection may be
made. We use four such criteria:
1. Percentage of total variance of TEC described by solar activity proxy should be the largest one.
2. The standard error of trend slope/coefficient D should be the smallest one.
3. Percentage of total variance of TEC residuals ($TEC_{obs}$ – $TEC_{mod}$) described by trend with the given solar proxy
should be the largest one.
4. The average of absolute values of differences between observed and model (with solar proxy) TEC (TEC
residuals) should be the smallest one.

**Table 1.** Global TEC JPL-35, 1994-2014, the fulfillment of criteria of selection of the optimum solar activity
proxy. $R^2$solar - percentage of total variance of TEC described by solar activity proxy. Slope D and its standard
error – trend coefficient. $R^2$trend - Percentage of total variance of TEC residuals ($TEC_{obs}$ – $TEC_{mod}$) described by
long-term trend. dTEC - The average of absolute values of differences between observed and model (with solar
proxy) TEC (TEC residuals).

|  | F10.7 | Fα | Mg II | sunspots | F30 | He II |
|---|---|---|---|---|---|---|
| $R^2$solar | 99% | 99% | 99% | 99% | 99% | 99% |
| Slope D (TECU/yr) | -0.048 ±0.025 | -0.060 ±0.026 | -0.067 ±0.028 | 0.012 ±0.032 | -0.108 ±0.024 | 0.100 ±0.050 |
| $R^2$trend | 0.16 | 0.21 | 0.23 | 0.01 | 0.52 | 0.21 |
| dTEC | 0.51 | 0.55 | 0.69 | 0.73 | 0.44 | 0.74 |


Table 1 show how these criteria are fulfilled for all six solar activity proxies used. The first row presents the
percentage of total variance of TEC described by individual solar activity proxies. These percentages are equal,
99%, for all solar activity proxies, thus this criterion does not help to select the optimal proxy. However, 99%





confirms that the linear equation (1) may be used, that it is not oversimplification. The second row show the
trend slope/coefficients and, more important, their standard errors. The smallest standard error (even though with
the highest trend slope) is for F30 but those for F10.7, Fα and Mg II differ very little. However, this criterion
disqualifies He II. The third row brings information about the percentage of total variance of TEC residuals
described by trend with individual solar activity proxies. This criterion clearly and very much favors F30
(percentage for F30 is more than twice as large as for all other solar activity proxies) and evidently disqualifies
sunspot numbers. The fourth criterion shown on the fourth row, the average of absolute values of differences
between observed and model TEC, again supports F30 as the optimum solar activity proxy. Summing up, we
may say that F30 is the optimum solar activity proxy for studying long-term trends of TEC based on yearly
values. This result is not surprising, because F30 is also the optimum solar activity proxy for foF2 as discussed in
Introduction and the F2 layer forms very substantial contribution to TEC.


**4  Long-term trends in TEC**

Since long-term trends in foF2 are most pronounced around noon (e.g., Danilov, 2015) and the F2 region
represents very important contribution to TEC, we focus on TEC trends around noon (10-14 LT). They are
calculated using equations (1) and (2) and solar activity proxy F30. These trends are presented in Figs. 1-3 in the
form of meridional profiles of trends separated by $30^o$ in longitude. All three Figures reveal a similar general
latitudinal pattern. At higher latitudes ($\varphi > 30^o$, for Fig. 3 $\varphi > 20^o$) at both hemispheres the trends are weak, close
to no trend, and dominantly insignificant except for the southern very high latitudes, which display a larger
negative trend; all longitudinal belts provide similar pattern. At lower latitudes the pattern is clearly different.
Strong negative trends occur for longitudinal belts 0-60$^o$E and 180-240$^o$E. On the other hand, longitudinal belts
90-150$^o$E and 270-330$^o$E reveal the same lower latitude pattern as higher latitude pattern, weak or no trends.

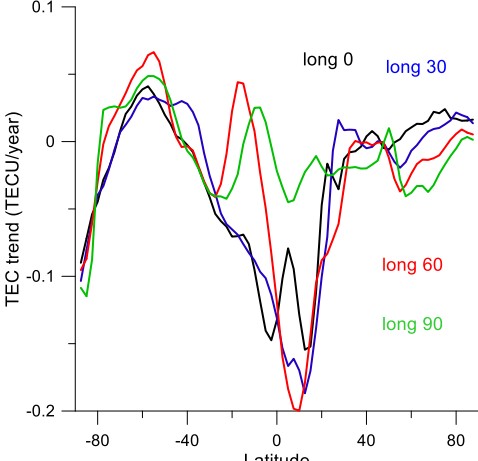


Fig. 1. Latitudinal dependence of TEC trends (TECU/year) for longitudinal belts centered at 0$^o$, 30$^o$, 60$^o$ and 90$^o$,
latitudes 87.5$^o$S-87.5 N.



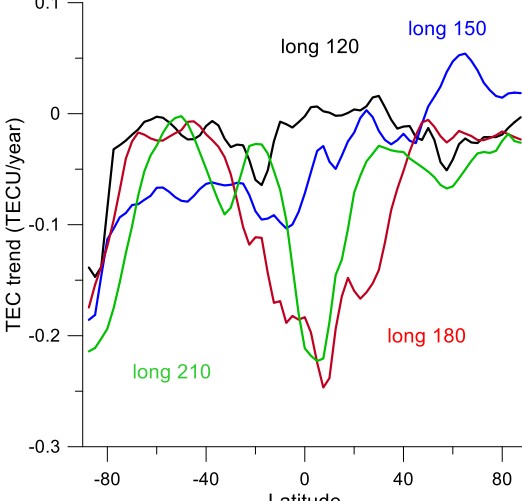

Fig. 2. Latitudinal dependence of TEC trends (TECU/year) for longitudinal belts centered at 120°, 150°, 180° and 210°, latitudes 87.5°S-87.5 N.

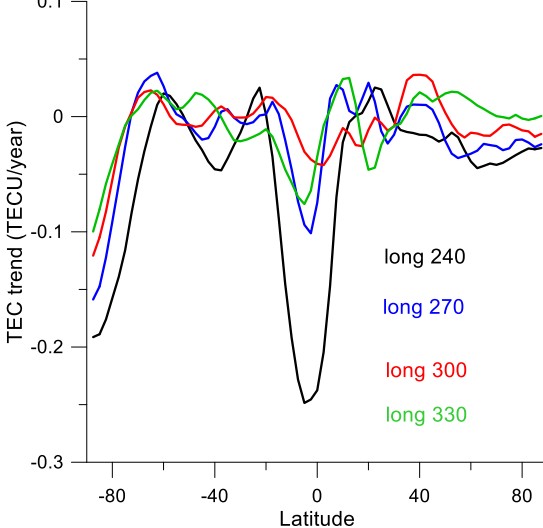

Fig. 3. Latitudinal dependence of TEC trends (TECU/year) for longitudinal belts centered at 240°, 270°, 300° and 330°, latitudes 87.5°S-87.5 N.

Important compound of trend investigations is statistical significance of results. The statistical significance of TEC trends is predominantly low. Trends with significance higher than two standard deviations (2 σ) occur for all profiles at southern very high latitudes (in average 80-87.5°S). Trend profiles with large low-latitude trends are significant at the 2 σ level typically between 20°N and 20°S, whereas profiles with weak trends are significant only in the vicinity of equator and for some profiles only. Profiles with weak low-latitude trends are mostly statistically significant at the 2 σ level also at northern higher middle latitudes (typically 50°-65°N). TEC



trends appear to be statistically significant at southern very high latitudes ($\varphi \geq 80^{o}$S), however these latitudes
suffer with low density of data. All other parts of trend profiles reveal lower statistical significance, many of
them even lower than 1 σ. One reason for so low significance of linear trend might be change of trend during the
analyzed period. To check this possibility, Fig. 4 shows temporal evolution of TEC trends in terms of TEC
residuals ΔTEC at 30$^{o}$E for latitudes with the strongest (12.5$^{o}$N) and weakest (40$^{o}$N) trends. 40$^{o}$N clearly reveals
no change of trend and also 12.5$^{o}$N does not show an evident change of linear trend. However, Fig. 4 displays
large year-to-year variability of ΔTEC; with such a large variability to get trends with sufficient statistical
significance requires for most of trend values longer data sets. In this sense our results might be considered
preliminary except for clear dominance of negative trends and a clear division of trends at low latitudes into four
groups of strong and weak trends.

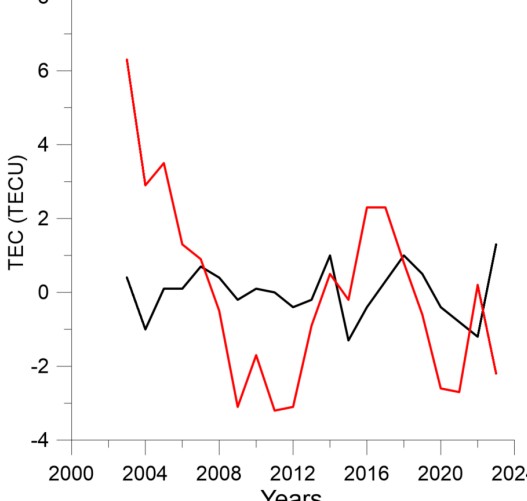


Fig. 4. Evolution of TEC residuals at 30$^{o}$E, 40$^{o}$N (black, no trend) and 12.5$^{o}$N (red, the largest trend).

Andima et al. (2019) analyzed TEC trends for equatorial station Malindi in Kenya; our negative trend value for
this region is within the range of their trend values. More positive/less negative trends of global TEC by Lean et
al. (2011, 2016) are explained by the use of TEC data prior to 2002 without any correction. This data problem
was unknown at the time of publication of Lean's et al. results; it was detected first by Emmert et al. (2017).

As concerns model simulations of trends in TEC, our global TEC trend -0.108±0.024 TECU/year (Table 1)
calculated with F30 is somewhat higher than the trend simulated by Cnossen (2020, Table 1), which reached
values between -0.060±0.012 and -0.024±0.008 TECU/year, but trend calculated with F10.7 (-0.048±0.025),
which was used by Cnossen (2020), are within the range of Cnossen (2020) trend values. McInerney et al. (2024)
used model WACCM-X to calculate TEC trends. They obtained for March and June, 1920-2010, zonal means,
negative trends of various magnitude at all latitudes. Thus our TEC trends at least qualitatively agree with the
trends from model simulations.



Why are low-latitude TEC trends separated into two longitudinally-separated groups of strong and weak trends?
Secular change of Earth's magnetic field does not seem to be responsible for the observed longitudinal structure
of the low latitude TEC trends, because it has pronounced impact on the low latitude ionospheric F2-region
trends in the 270-330°E belt (Qian et al., 2021), where TEC trends are weak. If the TEC trends shown in global
geographic coordinates are re-binned into the geomagnetic grid, this outcome will not change significantly.
Another possibility could be the effect of non-migrating tides. There is well-known effect of the DE3 non-
migrating tide on the low-latitude/equatorial ionosphere but it produces longitudinal structure with wavenumber
4, whereas TEC trends display longitudinal structure with the zonal wavenumber 2 at low/equatorial latitudes.
This problem requires more detailed study, which is out of the topic of this paper; it will be treated in future
investigations.


**5 Conclusions**

TEC is an important parameter for propagation and applications of GNSS/GPS signals. Despite this fact the
long-term trends in TEC have been studied a little only. Altogether five papers dealt with trends in observed
TEC until now (Lean et al., 2011, 2016; Lastovicka, 2013; Lastovicka et al., 2017; Andima et al., 2019) and their
results are not mutually consistent. The results of this work may be summarized as follows:
1. The TEC trends are mostly statistically insignificant at the 2 σ level, even though in some latitudinal-
longitudinal regions they are statistically significant. This means that only gross features, not fine
details, may be considered reliable. Longer data series is required for getting finer structure of TEC
trends.
2. The optimum solar activity proxy for investigating long-term trends in TEC is F30, not F10.7, Mg II or
sunspot numbers. This is consistent with F30 being the optimum solar proxy for foF2 trends
(Lastovicka and Buresova, 2023).
3. The long-term TEC trends are very predominantly negative; all statistically significant trends are
negative.
4. TEC trends reveal a clear zonal wavenumber 2 longitudinal structure in low/equatorial latitudes with
strong negative trends in belts 0-60°E and 180-240°E and weak trends in 90-150°E and 270-330°E.
Future investigations will focus on analysis of longer data series and on search for explanation of longitudinal
structure of TEC trends at low/equatorial latitudes.


*Data availability*.
Data used in this study are publicly available on the following websites:
Solar activity indices were taken from:
F10.7 (observed) - https://lasp.colorado.edu/lisird/data/noaa_radio_flux/,
F30 - https://solar.nro.nao.ac.jp/norp/data/daily/,
Lyman-α - https://lasp.colorado.edu/data/timed_see/composite_lya/version3/,
Mg II - http://www.iup.uni-bremen.de/UVSAT/Datasets/mgii,



sunspot numbers were taken from https://sidc.be/silso/datafiles,
He II - from the SOLID project database: https://projects.pmodwrc.ch/solid-
visualization/makeover/index.php?type=proxy&waveStart=215&waveEnd=215&dateStart=1970-01-
01&dateEnd=2014-12-31, with the option: Proxies > Data selections > He II > Download.
Global TEC data were taken from Emmert et al. (2017), supporting information, Data Set S1].
UPC TEC data were taken from https://cddis.nasa.gov/archive/gnss/products/ionex/2023/.

*Author contributions.*
J.L.: Conceptualization, analysis of global TEC data. J.U.: Data mining and analysis of UPC TEC data. Both:
Writing of manuscript.

*Competing interests.* The contact author has declared that none of the authors has any competing interests.

*Acknowledgements:* We acknowledge work of all who contributed to production of ionospheric and solar data
used in this article.

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
