# Peer review of "Global long-term trends in the total electron content"

_EGUsphere, 2024_

## Referee Comment (RC1)

**Reviewer comments to the paper by Urbar and Lastovicka**

The paper considers long-term trends in the total electron content TEC and their spatial distribution. A brief description of the TEC data sources and previous attempts to derive trends in TEC is presented in the Introduction.

Section 2 describes the TEC data used for the analysis. Even though the long-term trends in TEC are a final goal of the study, the authors use homogeneous series JPL-35 of global TEC data for 1994-2014 to find the best solar proxy for revealing TEC trends. The UPC TEC maps for 2003-2023 are used to reveal trends in TEC. Yearly averages based on monthly medians are analyzed for the near-noon hours (10-14 LT). Special steps are undertaken to have the data for various longitudinal belts compared in the same local time.

Usual method of ionospheric trend derivation is applied and very briefly described in Section 2.

The authors formulate four criteria for selecting the best solar proxy to derive trends in TEC and apply them to six solar proxies. The results are presented in Table 1. Although all four criteria are fulfilled for all six proxies considered, the authors select the F30 proxy as the best for further searches of the trends. Their choices coincides with results of similar studies on trends in the F2-layer parameters.

The results of revealing the TEC trends are presented in Section 4 in Figs. 1-3 in the form of latitudinal profiles of the trends for 12 longitudinal belts. In all tree figures with these results, an interesting pattern is seen: a strong increase in the magnitude of the negative trends is seen near the equator and at very high latitudes of the Northern Hemisphere.

The authors consider in detail features of each group of profiles. In particular, they describe a statistical significance of all profiles in Figs. 1-3. An important detail is that the derived trends have overall a low statistical significance except for strong enough trends in the 20N-20S belt and at high latitudes.

Figure 4 is aimed to show time variations of the trends for 2000-2024. It demonstrates that the ΔTEC values oscillate near zero at 40 N and provide some changes at 12.5 N.

Comparing their mean trend of -0.108 TECU/year with the results of model evaluations, the authors note that it is slightly higher than the model one.

An interesting feature of the derived TEC trends is that they reveal a longitudinal structure at equatorial latitudes with strong negative trends at 0-60E and 180-240E and weak trends at 90-150E and 270-330 E.

I consider the paper under review as a substantial input into our understanding of the very important problem of long-term trends in the F2-region parameters. I recommend publishing the paper with a minor revision.

My only critical comment is as follows.

I am slightly confused by Fig. 4. It is written in the text: "Fig. 4 shows temporal evolution of TEC trends in terms of TEC residuals ΔTEC…". I have found no indication in the paper on what is ΔTEC. "Residuals" of what with what? What is shown at the ordinate in Fig. 4? It is shown as TEC in TECU, but the values are typical for some difference. Is it really ΔTEC? I hope it could be easily corrected and made more understandable.

---

## Author Response (AR1)

1. Around lines 176, the authors discussed the global TEC trends shown in Table 1 and compared them with previous simulations. However, I suppose that the Table results are based on JPL-35 1994-2014 dataset for optimal solar activity proxy selection purpose. But the main objective of the paper is investigating the global TEC trend from UPC TEC 2003-2023 dataset as emphasized in the text and shown in Figures 1-4.  I suggest comparing the trend results from UPC TEC 2003-2023 dataset with simulation results. I might be misunderstanding. Please clarify this point.

*We compare trends in global TEC with models; preferentially with JPL-35 TEC due to the period analyzed but also with UPC TEC. Cnossen (2020) model simulations were for global TEC trend over 1950-2015, spatial variations of trends were calculated for other parameters, not TEC.  McInerney et al. (2024) used WACCM-X and calculated TEC trends for March and June, 1920-2010, zonal means only. It is important to note this TEC only includes the electron density contribution below the WACCM-X model top. Thus period of data set of JPL-35(1994-2014) corresponds better to simulated periods than used UPC data set (2003-2023) but trends from both periods (even though different) reasonably agree with Cnossen (2020).Both TEC datasets support F30as the best solar proxy. Until now nobody tried to simulate longitudinal variation of trends in TEC for comparison with our results. We modulated sentence on lines 196-197 to "Thus our trends in global TEC at least qualitatively agree with the trends from model simulations".*

2. In lines 188-191, I personally think the DE3 effect on the longitudinal TEC variations should be removed in terms of such large scale (yearly) calculations, except that the DE3 effect itself has long-term trend. The authors can talk more details on this issue.

*We state in the paper that DE3 creates different wavenumber structure than our observations at low/equatorial latitudes, i.e. DE3 does not explain our observations. We mention possibility of some non-migrating tides as perhaps potential explanation, because at present we do not have explanation of the observed longitudinal variability of low-latitude TEC trends.*

3. If using UPC TEC 2003-2023 dataset to generate Table 1, can the authors draw the similar conclusions as JPL-35 1994-2014? I am just curious.

*We preferred JPL-35, because this dataset has guaranteed data homogeneity, among others due to stable number of stations. However, we made the same analysis also for global UPC TEC at noon. The results are presented in new Table 2 and related text, and they again support F30 as the optimum solar activity proxy. He II proxy had some data quality problems in recent years (e.g., Lastovicka and Buresova, 2024, https://doi.org/10.1029/2022SW003359),therefore it is missing in Table 2.*

4. How and to what extend the longitudinal and latitudinal variations of TEC trend shown in Figures 1-3 are affected by the ground based GNSS receivers geographic distribution?  Since UPC TEC only used a little portion and sparse distributed ground receivers and the majority of TEC grids are actually the interpolated results.

*This is another reason why we have provided the trends only from year 2003, when the coverage by stations is better.*
*We do not think that quality of UPC data set is poor; then it would not be used as one of four data sets for constructing official GIC GIM maps. We do not expect much effect on*

*longitudinal and latitudinal distribution of trends (due to their character). Emmert et al. (2017) shows that in 2003-2014 the global UPS TEC and JPL-35 TEC vary in a similar way (keep constant difference).*

*We do not think that quality of UPC data set is poor; then it would not be used as one of four data sets for constructing official GIC GIM maps. They (UPC) are always listing which specific stations were involved in their computations. It was ~130-300 stations within the 2003-2023 interval. UPC estimates modeled independently for each station have not been interpolated. We do not expect much effect on longitudinal and latitudinal distribution of trends (due to their character). Emmert et al. (2017) shows that in 2003-2014 the global UPS TEC and JPL-35 TEC vary in a similar way (keep constant difference).*

*The lists of stations which were used for generation of UPC TEC for the each respective day seems to show that (except for some specific longer periods of some station outages) UPC TEC is providing reasonable coverage in most of the areas evaluated. Please note that we use wide latitudinal belts and so the interpolated bins might introduce problems only during intervals with missing measurements in anomaly areas like the Weddell sea. It was only relatively recently possible to considerably improve the TEC over oceans using the GNSS-RO so it will not be possible for many years to provide significantly better GNSS-based trend evaluations.*

*We agree that there are still problems concerning among others the interesting trends at the high Southern latitudes. There might be still some artifacts due to the interpolation of the results due to missing data. But for instance considering Antarctica, why the trend would change when the interpolation technique did not changed? UPC used are available inputs from 1998 from McMurdo/Scott base: mcm4 SCTB00ATA Latitude: -77.849, Longitude: 166.758 so at -80S the trend output should be still realistic (although missing some data in most of 2011-2015 and 2017-2018, 2022-2023). As more southern, e.g. the South Pole station data is not available within IGS at all, no other IGS IONEX product would perform considerably better than UPC we used for showing trends at high Southern latitudes. We are working on validation of the trends at possibly problematic regions in our related efforts. Anyway, in Conclusions we claim as reliable only the large dominance of negative trends and their longitudinal structure at low/equatorial latitudes, all other results including trend sin high southern latitudes need confirmation by longer data set analysis in future.*